

# *IGF-1* rs6218 polymorphisms modulate the susceptibility to age-related cataract

Xi Zou[1,2], Jun Zhang[2], Yong Wang[3], Dong Zhou[2], Guohua Deng[2] and Zhinan Liu[2]

[1] Changzhou Medical Center, Changzhou, China
[2] The Third People's Hospital of Changzhou, Changzhou, China
[3] Nantong First People's Hospital, Nantong, China

## ABSTRACT

**Background:** Single nucleotide polymorphisms (SNPs), as the most abundant form of DNA variation in the human genome, contribute to age-related cataracts (ARC) development. Apoptosis of lens epithelial cells (LECs) is closely related to ARC formation. Insulin-like growth factor 1 (IGF1) contributes to cell apoptosis regulation. Moreover, IGF1 was indicated to exhibit a close association with cataract formation. Afterward, an investigation was conducted to examine the correlation between polymorphisms in *IGF1* and the susceptibility to ARC.

**Methods:** The present investigation was a case-control study. Venous blood draws were collected from the participants for DNA genotyping. Lens capsule samples were collected to detect mRNA and apoptosis. TaqMan RT-PCR was used to detect *IGF1* polymorphism genotypes and qRT PCR was used to detect *IGF1* mRNA levels in LECs. LEC apoptosis was evaluated through flow cytometry. The chi-square test was used to compare differences between ARCs and controls of each SNP.

**Results:** We found that the G allele frequency in the *IGF1*-rs6218 was higher in the ARCs than in the controls. Furthermore, it was observed that the rs6218 GG genotype exhibited a positive correlation to elevated levels of *IGF1* mRNA in LECs. The *IGF1* mRNA in the LECs and the apoptosis of LECs in nuclear type of ARCs (ARNC) was higher than the controls.

**Conclusion:** The susceptibility to ARC was related to *IGF1*-rs6218 polymorphism, and this polymorphism is associated with *IGF1* expression at the mRNA level. Moreover, apoptosis in LECs of ARNCs was found to be increased.

## INTRODUCTION

Age-related cataract (ARC) is the transformation of the crystalline lens from transparent to cloudy, it is a multifactorial disorder standing as the primary contributor to blindness globally, especially in middle-income and low-income countries (*Cicinelli et al., 2023*). Although surgery can restore vision in most patients, the procedure itself has some risks and complications, which is a heavy economic burden for patients in developing countries and may lead to surgical complications, even irreversible blindness (*Bhandari & Chew, 2023*; *Karesvuo et al., 2022*). The risk of ARC was found to be affected by several genetic

Corresponding authors
Guohua Deng,
13584379133@163.com
Zhinan Liu, 20018542@qq.com

and environmental factors, including oxidative damage and metabolic disorders, among others (*Park & Lee, 2015*; *Tang et al., 2018*). However, the definite pathogeny of ARC remains incompletely understood.

Genetic variations, especially single nucleotide polymorphisms (SNPs), participate significantly in ARC development, SNPs are the most abundant form of DNA variation in the human genome, they can exist at any location of a gene, including intron, coding and untranslated region (*Zou et al., 2018*). Some of our previous genes and SNPs were associated with ARC: *LSS*-rs2968 A allele might play a role in the formation and development of nuclear type of ARC risk (*Zou et al., 2020*); the *NEIL2*-rs4639 T allele was strongly associated with a protective role in ARCs (*Kang et al., 2019*); the *XPC*-rs2229090 C allele was associated with ARNC risk (*Zou et al., 2018*).

Lens epithelial cells (LECs) are the primary site of metabolic activity within the lens (*Wang et al., 2017*). Apoptosis of LECs is the cellular basis of cataract formation (*Zheng et al., 2018*). LECs apoptosis leads to loss of homeostasis and accumulation of crystallin proteins, which then leads to the formation of cataracts (*Qi et al., 2022*). Low expression of MSH3 leads to apoptosis of LECs, which may lead to the occurrence of ARC (*Chen et al., 2022*).

The insulin-like growth factor (IGF) system is involved to a great extent in governing multiple biological processes, including metabolic function, cellular development, cell proliferation, differentiation, and apoptosis (*Tien et al., 2017*; *Willems et al., 2016*). This IGF system comprises IGF-1/2 besides their cell surface receptors (IGF-1R/2R) and six specific IGF-binding proteins (IGFBPs), IGF-1R participates in gene expression regulation by forming transcriptional complexes, modifying the activity of chromatin remodeling proteins, and participating in DNA damage tolerance mechanisms. IGF-1 is a small protein that binds to IGFBPs in circulation and stimulates IGF-1R upon release, which undergoes self phosphorylation (*Poreba & Durzynska, 2020*; *Cao et al., 2016*). IGF1 can activate their receptors and trigger the phosphorylation of IGF1R itself and downstream signal transduction, including PI3K/AKT and Ras/Raf/ERK signal pathways, inhibit cell proliferation and promote cell apoptosis (*Cao et al., 2020*).

Study shows SNPs of *IGF1R* are related to eye diseases (*Chiu et al., 2011*; *Cui et al., 2020*). *IGF1R*-rs2872060 revealed a significant association with advanced AMD (*Chiu et al., 2011*), *IGF1R*-rs1546713 may affect susceptibility to ARCs (*Cui et al., 2020*). Moreover, IGF1 was indicated to exhibit a close association with cataract formation (*Civil et al., 2000*; *Papier et al., 2022*; *Wang et al., 2022*). IGF1 might decrease rats' content of α-crystallin made in the lens Fibre cells, which leads to rat lens opacification and cataract formation (*Civil et al., 2000*). IGF1 promoted cataract formation might by promoting the epithelial-mesenchymal transformation of LECs (*Wang et al., 2022*). However, whether SNPs of *IGF1* were related to ARC is still not clear.

# MATERIALS AND METHODS

## Study participants

The study was granted approval from the Ethics Committee of Changzhou Third People's Hospital (ethical approval number: 2021012) and followed the Declaration of Helsinki.

**Table 1 Demographic information of epidemiologic participants.**

| Variable | $n$ | Age (Mean ± SD) | $p$ | sex | | $\chi^2$ | $p$ |
|---|---|---|---|---|---|---|---|
| | | | | Male (%) | Female (%) | | |
| Controls | 685 | 69.54 ± 5.35 | | 294 (42.92) | 391 (57.08) | | |
| ARCs | 716 | 69.59 ± 5.39 | 0.211 | 281 (39.25) | 435 (60.75) | 0.014 | 0.174 |
| C | 377 | 70.1 ± 5.64 | 0.451 | 167 (44.4) | 210 (55.6) | 0.052 | 0.524 |
| N | 223 | 68.9 ± 5.86 | 0.121 | 103 (46.4) | 120 (53.6) | 0.162 | 0.371 |
| PSC | 48 | 68.2 ± 4.16 | 0.192 | 22 (45.8) | 26 (54.2) | 0.026 | 0.491 |
| M | 68 | 70.1 ± 6.14 | 0.514 | 32 (47.1) | 36 (52.9) | 0.005 | 0.501 |

Note:
C, cortical cataract; N, nuclear cataract; PSC, posterior sub capsular cataract; M, mixed cataract.

The participants were aware of the study's aim and subsequently provided their informed consent by signing the appropriate documentation.

All participants underwent a comprehensive ophthalmic assessment, which included examinations of visual acuity, lens using a slit lamp biomicroscope under transient and side illumination after mydriasis, and ophthalmoscopic. The classification of ARC based on the opacity region of the lens includes four subtypes: cortical cataract (CC), nuclear cataract (NC), posterior sub-capsular cataract (PSC), and mixed cataract (MC) (*Klein, Klein & Linton, 1992*). The diagnosis and grading of lens opacities were conducted following the Lens Opacities Classification System III (LOCS III) (*Feng et al., 2022*).

This study adopts a case-control design, involving cases and controls from a population based epidemiologic cohort of the Jiangsu Eye Study located in Jiangsu Eye Study in Qidong country. In addition, individuals who served as controls and were matched with the experimental terms of age and sex and had transparent lenses were selected from the same communities. The geographical region under investigation exhibits a relatively stable and ethnically homogenous population. The participants were individuals sharing no familial relationships and self-identified as belonging to the Han Chinese ethnic group. Consequently, 716 ARC patients (CC = 377, NC = 223, PSC = 48, MC = 68) and 685 controls were included (Table 1). The inclusion/exclusion details of the case-control design followed the previous study (*Kang et al., 2019*).

In addition, we included additional 20 ARNC patients and 20 age-, sex- and ethnically matched controls from inpatients in our hospital from Jan 2019 to Dec 2021 (Table 2) to collect not only venous blood but also matched lens anterior capsule samples. Venous blood is used to detect its genotype, while lens anterior capsule samples were used to measure the mRNA levels of LECs and detect their apoptosis. The lens anterior capsule samples from ARC patients were obtained through anterior continuous circular capsulorhexis in phacoemulsification surgery. The lens anterior capsule samples from transparent lenses were acquired from patients who received lens extraction as part of vitrectomy procedures. The study excluded participants with lens trauma, diabetes, uveitis, glaucoma, and high myopia (>6D).

**Table 2 The grade of lens opacity and genetype of hospital participants.**

| Controls | | | | | ARNCs | | | | |
|---|---|---|---|---|---|---|---|---|---|
| Samples | Age (y) | Sex | LOCSIII | Genotype | Samples | Age (y) | Sex | LOCSIII | Genotype |
| No. 1 | 65 | Female | N0C1P0 | AA | No. 1 | 58 | Female | N3C0P0 | GG |
| No. 2 | 57 | Male | N0C0P0 | AG | No. 2 | 58 | Female | N5C0P0 | AA |
| No. 3 | 53 | Female | N0C0P1 | AA | No. 3 | 75 | Male | N4C0P0 | AG |
| No. 4 | 65 | Female | N0C1P0 | GG | No. 4 | 75 | Male | N3C0P0 | AA |
| No. 5 | 57 | Male | N0C0P0 | AA | No. 5 | 55 | Male | N5C0P0 | AA |
| No. 6 | 73 | Male | N0C0P1 | AA | No. 6 | 72 | Male | N3C0P0 | AA |
| No. 7 | 78 | Male | N0C1P0 | AA | No. 7 | 62 | Female | N3C0P0 | GG |
| No. 8 | 69 | Male | N0C0P0 | AG | No. 8 | 72 | Female | N4C0P0 | AA |
| No. 9 | 60 | Female | N0C1P0 | AA | No. 9 | 61 | Female | N3C0P0 | AG |
| No. 10 | 54 | Male | N1C0P0 | GG | No. 10 | 65 | Male | N3C0P0 | AA |
| No. 11 | 73 | Female | N0C0P1 | AG | No. 11 | 72 | Male | N5C0P0 | GG |
| No. 12 | 71 | Male | N1C0P0 | AA | No. 12 | 60 | Female | N4C0P0 | GG |
| No. 13 | 56 | Female | N1C1P0 | GG | No. 13 | 65 | Male | N3C0P0 | AA |
| No. 14 | 62 | Male | N0C0P0 | AG | No. 14 | 64 | Male | N3C0P0 | AA |
| No. 15 | 63 | Female | N0C0P1 | GG | No. 15 | 71 | Female | N3C0P0 | AG |
| No. 16 | 68 | Male | N0C0P0 | AA | No. 16 | 76 | Female | N3C0P0 | GG |
| No. 17 | 66 | Male | N0C0P0 | AA | No. 17 | 72 | Female | N5C0P0 | AA |
| No. 18 | 66 | Male | N0C1P0 | AG | No. 18 | 55 | Male | N3C0P0 | GG |
| No. 19 | 65 | Female | N0C0P0 | AA | No. 19 | 65 | Male | N4C0P0 | AA |
| No. 20 | 60 | Male | N0C0P0 | GG | No. 20 | 62 | Female | N3C0P0 | AG |

**Table 3 Descriptive information and statistics for seven SNPs candidate of *IGF1* gene.**

| SNPs | Chromosomal lacation | Nucleotide change | MAF | Location |
|---|---|---|---|---|
| rs6218 | 12:102,399,855 | A>G | 0.35 | 3'-UTR |
| rs5742714 | 12:102,396,074 | C>G | 0.14 | 3'-UTR |
| rs2288377 | 12:102,480,984 | A>T | 0.39 | Intron |
| rs35767 | 12:102,481,791 | C>T | 0.34 | Intron |
| rs5742612 | 12:102,481,086 | A>G | 0.37 | Intron |
| rs12579108 | 12:102,396,074 | C>G | 0.14 | 3'-UTR |
| rs12579077 | 12:102,483,308 | C>G | 0.39 | Intron |

**Note:**
MAF, minor allele frequency in Chinese population.

## Selection of SNPs

Haplotype-tagging SNPs of genes were chosen by conducting a search in NCBI dbSNP (https://www.ncbi.nlm.nih.gov/snp) using Han Chinese data. The study included SNPs that had a MAF of >10% and were predicted to be potentially functional using the SNP function prediction program (https://snpinfo.niehs.nih.gov/snpinfo/snpfunc.html) while excluding SNPs that exhibited strong linkage disequilibrium (LD) with adjacent variants, as determined by $r^2$ threshold ≤0.80 (Table 3).

## DNA preparation and genotyping

The study extracted genomic DNA from the venous blood of all participants through the Qiagen Blood DNA Mini Kit (Qiagen, Valencia, CA, USA), following the protocols.

The SNP genotyping analysis was conducted using the TaqMan genotyping assay (Thermos Fisher, Foster City, CA, USA) per the guidelines as previously described in our published works (*Kang et al., 2019*; *Zou et al., 2018*, *2020*).

## RNA extracting and quantification of *IGF1* mRNA expression

Total RNA was isolated by Trizol reagent from LECs (Invitrogen, Carlsbad, CA, USA). Then cDNAs were performed by PrimeScript RT reagent Kit (TaKaRa, Dalian, China). The total RNA extracted in this experiment was detected using a UV spectrophotometer, and the UV absorption values (260/280) of the RNA were all between 1.9 and 2.1, indicating a high purity of the extracted RNA.

The study employed TaqMan gene expression assay probes (Thermos Fisher, Waltham, MA, USA) to quantify *IGF1* mRNA (assay ID: Hs01547656_m1). The housekeeping gene control utilized in this study was Human GAPDH (assay ID: Hs02786624_g1). The study also conducted real-time PCR analysis using the ABI StepOne plus real-time PCR system (Applied Biosystems, Foster City, CA, USA) as well as utilized the $2^{(-\Delta\Delta Ct)}$ algorithm for calculating the fold change of gene mRNA level.

## Annexin V/PI apoptosis detection

Apoptosis was measured by the Annexin V-FITC/PI apoptosis detection kitt (BD, Franklin Lakes, NJ, USA). LECs were seprarated from the lens anterior capsule by using trypsin. The LECs were stained cells through the per protocols. After staining 15 min, flow cytometry was employed for cell detection on a CytoFLEX system (Beckman Coulter, Brea, CA, USA).

## Statistical analysis

The study utilized a commercial statistical software program (Stata 8.0) for performing the statistical analyses, reporting the data as means ± SD. The t-test was utilized to conduct statistical comparisons between the average values of the two groups. The $\chi 2$ test was employed for evaluating the relationship between the allele frequencies of ARC patients and normal controls, different ARC subtypes, odds ratios (OR), and 95% confidence intervals (CI), and also for testing Hardy-Weinberg Equilibriums (HWE) of genotype distributions. Upon detecting any positive correlation during the initial allele analysis, a Bonferroni correction was applied. The results only presented the most significant model. The qRT-PCR assays were conducted in a minimum of three independent replicates. $p < 0.05$ is indicative of a significant difference.

# RESULTS

## The participant features for the association study

This study enrolled participants from the hospital and epidemiologic, and the general demographic details of the participants were listed in Tables 1 and 2, respectively. A total

**Table 4 Genetic associations between polymorphisms in *IGF1* and risk of ARC.**

| SNPs Major/Minor | Controls Major/Minor | ARCs Major/Minor | $\chi^2$ | p/pa | OR (95% CI) |
|---|---|---|---|---|---|
| rs6218 A/G | 901 (65.8)/469 (34.1) | 865 (60.4)/567 (39.6) | 8.636 | **0.003/0.021** | 1.259 [1.080–1.469] |
| rs5742714 C/G | 913 (66.6)/457 (33.4) | 987 (68.9)/445 (31.1) | 1.671 | 0.210 | 0.901 [0.769–1.056] |
| rs2288377 A/T | 1,008 (73.6)/362 (26.4) | 1,062 (74.2)/370 (25.8) | 0.124 | 0.731 | 0.968 [0.819–1.148] |
| rs35767 C/T | 898 (65.5)/472 (34.5) | 961 (67.1)/471 (32.9) | 0.765 | 0.401 | 0.932 [0.917–1.254] |
| rs5742612 A/G | 984 (71.8)/386 (28.2) | 1,034 (72.2)/398 (27.8) | 0.051 | 0.833 | 0.981 [0.832–1.157] |
| rs12579108 C/G | 961 (70.1)/409 (29.9) | 1,025 (71.6)/407 (28.4) | 0.696 | 0.406 | 0.993 [0.793–1.098] |
| rs12579077 C/G | 985 (71.9)/385 (28.1) | 1,027 (71.7)/405 (28.3) | 0.011 | 0.933 | 1.009 [0.856–1.189] |

**Note:**

pa, p value after Bonferroni correction. Bold indicates $p < 0.05$.

of 716 patients with ARCs were included in the study, among which the numbers of CC, NC, PSC, and MC were 377, 223, 48, and 68, respectively. There were 685 age-, sex- and ethnically matched healthy control subjects. The general demographic details of the study participants are summarized in Table 1. Among the hospitalized population, there were 20 ARNC patients (AA genotype = 10; AG genotype = 4; GG genotype = 6) and 20 controls (AA genotype = 10; AG genotype = 5; GG genotype = 5) (Table 2). The average age is 62.6 ± 8.24 years in controls and 64.3 ± 6.42 years in ARNC patients. The ratio of sex is 0.55 in ARNC patients and 0.5 in controls. There is no difference between age and sex.

## Bioinformatics selection of candidate SNPs

The SNPs of *IGF1* were chosen for genotyping, listing their basic features in Table 3. Among them, rs6218, rs5742714, rs12579108 were located in the 3′-UTR region; rs2288377, rs35767, rs5742612, rs12579077 were located in the intron region of the *IGF1* gene.

## SNPs and ARC risk correlation

Between the 7 SNPs, the *IGF1*-rs6218 allele frequency of ARCs differed significantly from controls ($p = 0.003$). This significance remained even after applying multiple comparison corrections (Bonferroni correction) ($p = 0.021$). The findings indicated that the minor allele G frequency in the *IGF1*-rs6218 was higher in the ARCs than in the controls (Table 4). Subsequently, stratification analysis was further conducted to investigate the SNP contribution in ARC subtypes, revealing that the minor allele frequency of *IGF1*-rs6218 was only higher in the nuclear type of ARCs (ARNCs) than in the controls ($p = 0.021$) (Table 5).

**Table 5 Genetic associations between the *IGF1*-rs6218 polymorphism and various types of ARC.**

| Gene/SNP | Genotype | Controls, *n* (%) | C, *n* (%) | N, *n* (%) | PSC, *n* (%) | M, *n* (%) |
|---|---|---|---|---|---|---|
| *IGF1*/rs6218 | A | 901 (65.8) | 468 (54.1) | 258 (29.8) | 60 (6.9) | 79 (9.1) |
| | G | 469 (34.1) | 286 (50.4) | 188 (33.2) | 36 (6.3) | 57 (10.1) |
| $\chi^2$ | | | 2.902 | 9.138 | 0.424 | 3.209 |
| *P* | | | 0.091 | **0.003** | 0.507 | 0.089 |
| OR (95% CI) | | | 0.852 [0.708–1.025] | 1.400 [1.125–1.742] | 1.153 [0.751–1.768] | 1.386 [0.969–1.984] |

**Note:**
C, cortical cataract; N, nuclear cataract; PSC, posterior sub capsular cataract; M, mixed cataract. Bold indicates $p < 0.05$.

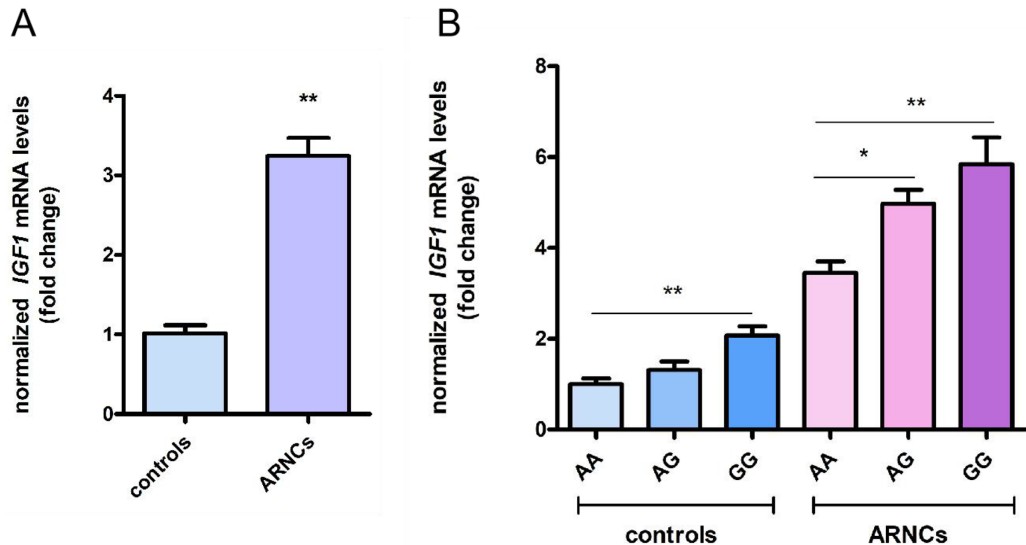

**Figure 1 Levels of *IGF1* mRNA expression of LECs in anterior capsules.** (A) *IGF1* mRNA levels were higher in ARNCs than in the controls (20 ARNC patients and 20 age-, sex- and ethnically matched controls). (B) *IGF1* mRNA levels were higher in the GG or AG group than the AA group in the ARNCs (AA genotype = 10; AG genotype = 4; GG genotype = 6). *IGF1* mRNA levels were higher in the GG than the AA group in the controls (AA genotype = 10; AG genotype = 5; GG genotype = 5).*$p < 0.05$. **$p < 0.01$. The qRT-PCR were repeated three times independently. Data were presented as means ± SD.

### rs6218 impacts on the mRNA levels of *IGF1* in biopsy samples

*IGF1* mRNA expression was higher in LECs of the ARNCs than in the controls (Fig. 1A). Additionally, *IGF1* mRNA expression of GG-genotype individuals was higher than AA-genotype individuals in both ARNCs and controls. In the ARNCs, *IGF1* mRNA levels were significantly higher in the GG or AG group than in the AA group. In the controls, *IGF1* mRNA levels were higher in the GG than in the AA group (*: $p < 0.05$; **: $p < 0.01$) (Fig. 1B).

### Apoptosis of LECs in biopsy samples

Apoptosis was a normal physiological process that orderly controls cell death to maintain stable homeostasis. The experimental results showed that cell apoptosis was present in both the ARNCs and the control group of LECs. However, The apoptosis of LECs in the

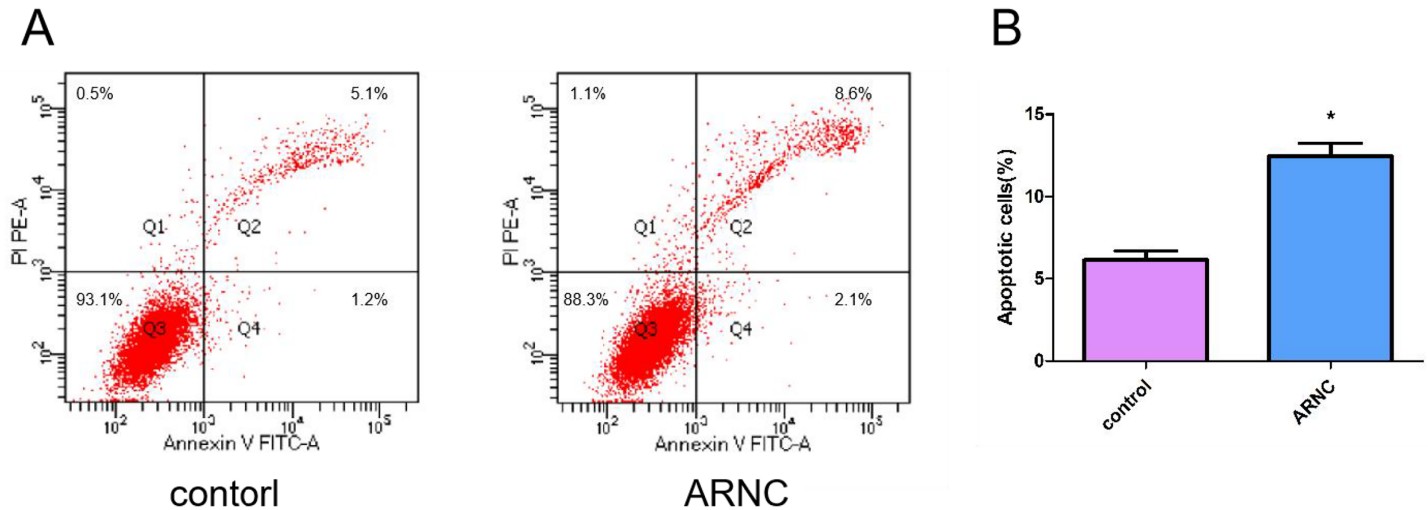

**Figure 2 Representative images of cell apoptosis.** (A) FCM analysis of the effect on cell apoptosis in LECs in anterior capsules of controls and ARNCs (20 ARNC patients and 20 age-, sex- and ethnically matched controls). (B) Quantitative data of (A). *$p < 0.05$.

anterior capsules of ARNCs was significantly higher than in the controls (Fig. 2). *: $p < 0.05$.

## DISCUSSION

Understanding the mechanism of ARC, which is recognized as the inaugural ocular disease causing blindness, is of utmost importance. SNPs contribute significantly to ARC development. Herein, Several genes and SNPs were revealed to be associated with ARCs (*Cui et al., 2020*; *Kang et al., 2019*; *Zou et al., 2018*, *2020*).

LEC apoptosis has a close association with ARC formation, oxidative stress-induced apoptosis of LECs is the main factor in the pathogenesis of ARC (*Tian et al., 2020*; *Zheng et al., 2018*). Our previous research reported a correlation between oxidative damage of LECs and the occurrence and development of ARC (*Kang et al., 2019*; *Zou et al., 2018*), but did not detect apoptosis. Studies suggested that the LEC apoptosis level in ARC patients was significantly higher than in healthy individuals (*Tian et al., 2020*). In this study, we found the apoptosis of LECs in the anterior capsules of ARNCs was higher than that of the controls.

IGF system was found to be involved in the apoptosis regulation process (*Willems et al., 2016*; *Tien et al., 2017*). Wherein, IGF1 can inhibit cell proliferation and promote cell apoptosis (*Cao et al., 2020*). In addition, IGF1 has a close association with cataract formation (*Civil et al., 2000*; *Papier et al., 2022*; *Wang et al., 2022*). Study found *IGF1R*-rs1546713 may affect susceptibility to ARCs (*Cui et al., 2020*). In our study, we found the minor allele G frequency in the *IGF1*-rs6218 was higher in the ARCs than in the controls. Further performed stratification analysis revealed that the minor allele G frequency of *IGF1*-rs6218 was only higher in ARNCs than in the controls. This indicates that *IGF1*-rs6218 G allele might play a role in the formation and development of ARNC risk in Chinese population. This study also found that *IGF1* mRNA expression was higher in the

LECs of the ARNCs compared to the controls. Furthermore, the mRNA expression of *IGF1* in GG-genotype individuals was higher than in AA-genotype individuals.

IGF1 can regulate cell apoptosis through targeted regulation by miRNA (*Wang et al., 2020*); rs6218 is located in the 3′-UTR region of the *IGF1* gene that is predominantly related to the binding of microRNAs (miRNAs). The binding of miRNA to target genes leads to mRNA degradation or post-transcriptional inhibition, thereby inhibiting gene expression (*Kang et al., 2019*; *Zheng et al., 2018*). Consequently, we postulated that rs6218 might potentially alter the binding energy between *IGF1* and miRNAs. Many miRNAs that can bind to *IGF1* have been discovered, but unfortunately, no miRNAs that can bind near the rs6218 site have been found yet through the online database (https://compbio.uthsc.edu/miRSNP/miRSNP_detail_all.php). The aforementioned results implicated that there are additional underlying mechanisms behind *IGF1* expression regulation in lens changing in LECs of ARNC patients. Nevertheless, this study had certain limitations; there are many reasons for the apoptosis of LECs, which may not be solely the result of *IGF1* upregulation. Further experiments are needed to detect apoptosis by enhancing or knocking out the *IGF1* gene *in vitro* experiments.

## CONCLUSION

*IGF1*-rs6218 G allele might play a role in the formation and development of ARNC risk in Chinese population, and this polymorphism is associated with *IGF1* expression at the mRNA level. *IGF1* mRNA expression was higher in the LECs of the ARNCs compared to the controls. In addition, the expression of *IGF1* mRNA in individuals carrying the G allele genotype was higher than that in individuals carrying the G allele genotype in both the ARNCs and controls. Moreover, apoptosis in LECs of ARNCs was found to be increased compared to the controls.

### Funding

This work was supported by the Scientific research project of Changzhou Medical Center of Nanjing Medical University (CZKYCMCB2022251); Science and Technology Project of Changzhou (CJ20220097; CE2022507); Project of Changzhou Health Commission (QN202129; ZD202120); Young Talent Development Plan of Changzhou Health Commission (CZQM2022017; CZQM2020091); Jiangsu Province Traditional Chinese Medicine Technology Development Project (MS2022078). The funders had no role in study design, data collection and analysis, decision to publish, or preparation of the manuscript.

### Grant Disclosures

The following grant information was disclosed by the authors:
Scientific research project of Changzhou Medical Center of Nanjing Medical University: CZKYCMCB2022251.
Science and Technology Project of Changzhou: CJ20220097, CE2022507.

Project of Changzhou Health Commission: QN202129, ZD202120.
Young Talent Development Plan of Changzhou Health Commission: CZQM2022017,
CZQM2020091.
Jiangsu Province Traditional Chinese Medicine Technology Development Project:
MS2022078.

## Competing Interests

The authors declare that they have no competing interests.

## Author Contributions

- Xi Zou conceived and designed the experiments, performed the experiments, prepared figures and/or tables, and approved the final draft.
- Jun Zhang performed the experiments, prepared figures and/or tables, and approved the final draft.
- Yong Wang conceived and designed the experiments, prepared figures and/or tables, and approved the final draft.
- Dong Zhou analyzed the data, prepared figures and/or tables, and approved the final draft.
- Guohua Deng conceived and designed the experiments, authored or reviewed drafts of the article, and approved the final draft.
- Zhinan Liu analyzed the data, prepared figures and/or tables, authored or reviewed drafts of the article, and approved the final draft.

## Human Ethics

The following information was supplied relating to ethical approvals (*i.e.*, approving body and any reference numbers):

The study was granted approval from the Ethics Committee of Changzhou Third People's Hospital.

## Data Availability

The original measurement values are available in Supplemental Files.

## Supplemental Information

Supplemental information for this article can be found online at http://dx.doi.org/10.7717/peerj.17220#supplemental-information.

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
