# Peer review of "IGF-1 rs6218 polymorphisms modulate the susceptibility to age-related cataract"

_PeerJ, doi:10.7717/peerj.17220_

## Round 0.1 · original submission · Major Revisions

Reviewers have identified several issues in your manuscript, please attend carefully to each and resubmit including the corresponding response letter, and remember to use the track changes tool in your new manuscript version.

**Language Note:** The review process has identified that the English language must be improved. PeerJ can provide language editing services - please contact us at [email protected] for pricing (be sure to provide your manuscript number and title). Alternatively, you should make your own arrangements to improve the language quality and provide details in your response letter. – PeerJ Staff

Reviewer 1 ·

Basic reporting

English used through the entire manuscript could be improved. A number of sentences are unclear or ambiguous.

Literature cited is sufficient to support the purpose of the work.

The struxture of the article must be improved. Results section is very limited regarding explanation and description of the particular results. This limits the understanding of the results.
The abstract lacks a conclusion section.
Figure legends are very plain. it should include additional information. For example, number of samples analyzed, technical repeats, etc.
Raw data is availabel

Experimental design

The research question is appropriate nd well defined.
Methods are insufficiently describes. No description of cataract tissue manipulation or specimen characteristics was provided (i.e., age of patients, time of specimen analysis after surgery). No information about how many samples were analyzed to obtain the mRNA expression and apoptosis data shown. It is not clear if comparative graphics on GG and AA genotypes are from cases and controls or from only a group.

Validity of the findings

The results are interesting, however given the limited data on technical procedures, it is not clear at this point the validity of them. The authors need to clearly state how many samples were analyzed to obtain their IGF1 mRNA expressión and LECs apoptosis data. Akso, they shoul expand the description of thechnicalmanipulations of the samples.

Additional comments

None

Reviewer 2 ·

Basic reporting

• English Language: needs significant improvement.
• Introduction and background: need elaboration.
• Structure confirms to PeerJ standards.
• Figures are relevant, high quality, but labels & description needs improvement.
• Raw data supplied.

Human participant/human tissue checks

• Have you checked the authors ethical approval statement? Does the study meet our article requirements?
Yes

• Has identifiable info been removed from all files?
Remove age criteria from the raw data file, you may add age range, but exact age is considered in breach of patients confidentiality.

• Were the experiments necessary and ethical?
Yes

Experimental design

• Original primary research within Scope of the journal.
• Research question well defined, relevant & meaningful.
• It is stated how the research fills an identified knowledge gap. Rigorous investigation performed to a high technical & ethical standard.
• Methods description needs improvement and more details.

Validity of the findings

• Study has significant impact and novelty.
• Most underlying data have been provided; they are robust, statistically sound, & controlled. I recommend including mRNA gels in figures.
• Conclusions need improvement.

Additional comments

The manuscript covers an elaborate and comprehensive study. The quantity and quality of experiments performed are commendable and the findings are novel and of significant impact. However, the authors are selling themselves short by not appropriately reporting their research. I recommend looking at other articles of similar approach as examples and/or using a professional editing service.

Page 4 Abstract:
Too short (168 words). Abstracts are usually 200 words minimum. Can be expanded in the background part since it needs more information.

Page 5 Title Page:
Corresponding author:
There are two corresponding authors?

Line 24: spelling mistake Backgroud, should be Background.

Introduction:
The logic of the introduction needs to be refined, in particular by stating the reasons for undertaking the study, such as the currently researched shortages, and the significance of this study. Also t about 220 words, the introduction is very short. Introductions need to be between 500-1000 words.
Suggested points than can be included in the introduction:
• IGF-1 pathway and its relationship to IGFBPs.
• Molecular mechanism of Age-related cataract (ARC).
• How is Lens epithelial cells (LECs) apoptosis related to ARC?
Methods:
• Study Participants: Please mention the period of sample collection, e.g. 2018-2020.
• Line 70: add the statement “ethical approval number 2021012”
• Line 105: change veinal blood to venous blood.
• Line 108 “as previously described in our published works.”: add reference(s).
• 116 Annexin V/PI Apoptosis Detection: needs rewriting for missing information and poor English language.
Results:
Try to explain more the results.
• Line 138: Elaborate: describe what is presented in the table and what is the most important point.
• Line 141: were all the SNPs alleles detected in your sample? Were all the SNPs in Hardy-Weinberg Equilibrium (HWE)?
• Line 150: “IGF1 mRNA expression of GG-genotype individuals was higher than AA- genotype individuals.” In both controls and patients?
• Line 152: IGF-1 font to italics.
• Line 155: elaborate.
Discussion:
Discuss in greater detail how your results fit in or not with the literature.
Compare what you did to what others did.
• Line 171: correct spelling “Fbre”
Table 1: subsets ARCs seem like additional groups; it is better to list them under ARC with increased indentation.
Table 2: change veinal to venous. Also this table may added to previous table.
Table 3: Change title: “The 7 SNPs selected from IGF1 gene” to more appropriate title, check literature for examples.
Table 4: check literature for examples on how to write caption for such table.
p values should be expressed as small p not capital P.
Table 5: caption needs to be edited.
Figure 1: add gel images.
Figure 2: I recommend improving title and caption. Also use small p for p value.

---

## Round 0.2 · accepted · Accept

Thanks for attending to all previous reviewers' concerns. One of the reviewers who had previously been really critical in his review now agrees with the changes made.
The current version is ready to be published.

Reviewer 1 ·

Basic reporting

The authors have addressed my queries appropriately. The manuscript quality has greatly enhanced.

Experimental design

The authors have addressed my queries appropriately. The manuscript quality has greatly enhanced.
Experimental procedures and results are now better presented and are more clear for readers understanding.

Validity of the findings

The authors have addressed my queries appropriately. The manuscript quality has greatly enhanced.
Experimental procedures and results are now better presented and are more clear for readers understanding.

Additional comments

None